# Comparative Transcriptomic Analysis Reveals Adaptation Mechanisms of Bean Bug *Riptortus pedestris* to Different Food Resources

**DOI:** 10.3390/insects14090739

**Published:** 2023-08-31

**Authors:** Ze-Long Zhang, Xiao-Jing Wang, Hai-Bin Lu, Hai-Jian Huang

**Affiliations:** State Key Laboratory for Managing Biotic and Chemical Threats to the Quality and Safety of Agro-Products, Key Laboratory of Biotechnology in Plant Protection of Ministry of Agriculture and Zhejiang Province, Institute of Plant Virology, Ningbo University, Ningbo 315211, China; zlzhang95@outlook.com (Z.-L.Z.); 2211130055@nbu.edu.cn (X.-J.W.); 2111074025@nbu.edu.cn (H.-B.L.)

**Keywords:** gut, transcriptomic, adaptation, *Riptortus pedestris*, food source

## Abstract

**Simple Summary:**

The bean bug *Riptortus pedestris* causes a significant decrease in soybean production and economic losses. To understand how these bugs adapt to different parts of the soybean plant, we studied their gut transcriptomic changes when feeding on soybean leaves, beans, and pods. We discovered that when they fed on pods and beans, there was a notable increase in the expression of digestive enzymes like cathepsins, serine proteases, and lipases. On the other hand, when the bugs consumed soybean leaves, detoxification enzymes, such as ABC transporters and 4-coumarate-CoA ligase, showed higher expression levels. These findings suggest that the dynamic regulation of digestive and detoxification enzymes enables bean bugs to successfully feed on various soybean tissues.

**Abstract:**

The bean bug, *Riptortus pedestris* (Hemiptera: Heteroptera), poses a significant threat to soybean production, resulting in substantial crop losses. Throughout the soybean cultivation period, these insects probe and suck on various parts of plants, including leaves, pods, and beans. However, the specific mechanisms by which they adapt to different food resources remain unknown. In this study, we conducted gut transcriptomic analyses of *R. pedestris* fed with soybean leaves, pods, and beans. A total of 798, 690, and 548 differently expressed genes (DEGs) were monitored in G—pod vs. G—leaf (comparison of insect feeding on pods and leaves), G—bean vs. G—leaf (comparison of insect feeding on beans and leaves), and G—pod vs. G—bean (comparison of insect feeding on pods and beans), respectively. When fed on pods and beans, there was a significant increase in the expression of digestive enzymes, particularly cathepsins, serine proteases, and lipases. Conversely, when soybean leaves were consumed, detoxification enzymes, such as ABC transporters and 4-coumarate-CoA ligase, exhibited higher expression. Our findings indicate that *R. pedestris* dynamically regulates different metabolic pathways to cope with varying food resources, which may contribute to the development of effective strategies for managing this pest.

## 1. Introduction

Under long-term coevolution with herbivores, plants have developed various defense mechanisms, such as toxic chemicals and physical barriers, to prevent damage of phytophagous insects [1,2]. Plant secondary metabolites like phenols, alkaloids, flavonoids, and terpenes have been shown to impede the growth and population development of insects [2,3,4]. These toxic metabolites are difficult to digest or are neurotoxic to insects [2,5]. In response, insects have evolved diverse strategies, including detoxification enzymes, such as cytochrome P450 (P450s) monooxygenases, carboxylesterases, and glutathione S-transferases (GSTs), to adapt to different host plants and detoxify phyto-toxins [6,7]. These degradation enzymes play a direct and/or indirect role in the breakdown of harmful plant metabolites and insecticides, enabling insects to successfully adapt to different host plants [7,8,9,10,11].

The bean bug *Riptortus pedestris* poses a substantial threat to soybean cultivation in Asia. Its remarkable ability to consume diverse soybean plant components—ranging from leaves to pods and beans—has significant implications for both crop quantity and quality [12,13,14]. The occurrence of the soybean *Zhengqing*, causing significant losses in soybean production, has been attributed to *R. pedestris* [15,16]. In the wild, *R. pedestris* exhibits an intriguing dietary behavior. Its presence in fields precedes the maturity of soybean beans, necessitating a period of sustenance from the tender leaves of nascent soybean seedlings [16]. This dietary adaptation underscores the dynamic variance in phytochemical compositions across distinct soybean tissues. Metabolomic analyses have elucidated the chemical tapestry of soybean leaves, pods, and beans. Leaves are enriched with compounds, such as flavonoids, phenolic acids, and alkaloids, which function as natural defense mechanisms against insect predation [17,18]. In contrast, beans are reservoirs of lipids and proteins, offering a nutritionally distinct profile [17,18]. Rahman and Lim found that *R. pedestris* exhibited reduced longevity and higher mortality when fed with pods and beans compared to beans alone [19]. Despite this knowledge, the mechanisms by which *R. pedestris* adapt to different hosts remain unclear.

Understanding the intricate mechanisms that govern the feeding preferences of *R. pedestris* across various plant parts holds profound significance, and it not only unravels the ecological intricacies of pest-host interactions but also offers a gateway to pragmatic agricultural solutions. For example, our recent study revealed key lipases linked to feeding and reproduction in *R. pedestris* [19]. These findings could guide us in using RNAi to control this harmful pest more effectively. As *R. pedestris* adaptation to varied diets is intricate, it is plausible that beyond lipases, other genes play pivotal roles in this process. Therefore, a comprehensive exploration of gene expression patterns using omics studies becomes imperative in unearthing the underlying mechanisms.

The gut is the main organ where digestion and detoxification occur, and it exerts an important influence on insect feeding on specific plants [20]. In this study, we performed gut transcriptomic analysis to investigate the physiological changes in *R. pedestris* when feeding on soybean leaves, pods, and beans. Differentially expressed genes (DEGs) were identified by comparing gene expression levels within each group. We found that digestive enzymes were important for bugs feeding on beans and pods, while the detoxification enzymes were responsible for bugs adapting to soybean leaves.

## 2. Materials and Methods

### 2.1. Insect and Plant

The *R. pedestris* was originally collected in a soybean field in Suzhou, Anhui, China in 2019 (33.7° N, 117.0° E), and reared on soybean (*Glycine max* L. Merr.) pods-containing plants in green house of Plant Virology Institute, Ningbo University. The soybean variety used in this study was *Wandou 27*. Both insects and plants were maintained at 26 ± 5 °C with a 16:8 (light:dark) photoperiod and humidity of 50 ± 5%.

### 2.2. Observation of Salivary Sheath

To investigate the impact of *R. pedestris* on soybeans, three branches of leaves, twenty beans, and twenty pods were put into net cages (25 cm × 25 cm × 25 cm) separately. Then, ten adults *R. pedestris* were released onto each net cage. The infested leaves and pods were collected five days post infestation. The healthy bean, leaf, and pod without *R. pedestris* infestation were used as controls. The morphology of each sample was photographed using a Canon EOS 80D camera (Canon Inc., Tokyo, Japan). The infested leaves and pods were carefully collected and cut into small pieces approximately 0.5 cm in width and 1 cm in length. The collected beans, pods, and leaves were then attached to stubs and subjected to an air-drying process at 37 °C in a baking oven. Once dried, the samples underwent gold-sputtering and were observed using an SEMTM4000 II plus microscope (Hitachi, Tokyo, Japan) [21].

### 2.3. Sample Collection and Transcriptome Sequencing

A group of forty 5th instar nymphs of *R. pedestris* were fed with pods, beans, and leaves for a duration of four days in separate net cages (25 cm × 25 cm × 25 cm). The insect guts were dissected in a phosphate-buffered saline (PBS) solution (137 mM NaCl, 2.68 mM KCl, 8.1 mM Na_2_HPO_4_ and 1.47 mM KH_2_PO_4_ at pH 7.4) with a pair of forceps under a stereomicroscope (Olympus SZX16; Tokyo, Japan). The dissected guts were washed in a clean PBS solution to avoid the contamination of other tissues. As it is difficult to remove the ingested food inside the lumen, we did not clean the gut lumen. Whole gut, including foregut, midgut, and hindgut, were collected for analysis. The collected samples were transferred into RNAiso plus (TaKaRa, Dalian, China) and kept on ice during the dissection period. Guts dissected from ten insects were pooled and regarded as one replicate. RNA was extracted according to the manufacturer’s protocol. The chloroform (Sango Biotechnology, Shanghai, China), isopropanol (Sango Biotechnology), and 75% ethanol (Sango Biotechnology) were used for RNA extraction. RNA integrity and quantity was assessed using the RNA Nano 6000 Assay Kit of the Bioanalyzer 2100 system (Agilent Technologies, Santa Clara, CA, USA). The RNA samples were sent to Novogene (Tianjin, China) for library construction and Illumina sequencing.

In detail, the library construction process involved several steps. Initially, poly (A) + RNA was purified from a pooled total RNA sample of 20 μg using oligo (dT) magnetic beads. Subsequently, fragmentation of the RNA was carried out in the presence of divalent cations at 94 °C for 5 min. N6 random primers were then employed for reverse transcription, resulting in the generation of double-stranded complementary DNA (cDNA). To prepare the cDNA library, end-repair and adaptor ligation were performed. The products were then subjected to 16 cycles of polymerase chain reaction (PCR) amplification with Phusion High-Fidelity DNA polymerase, Universal PCR primers, and Index (X) Primer according to the manufacturer’s protocol. Subsequently, samples were purified using a QIAquick PCR purification kit (Qiagen, Hilden, Germany). The prepared library was sequenced on an Illumina Nova6000 platform with 150-bp paired-end reads. For each treatment, cDNA library construction and Illumina sequencing were carried out with three biological replicates. The output data were submitted to the National Genomics Data Center under accession number: PRJCA019207.

The internal software Fastp version 0.23.1 [22] was used to obtain clean reads by removing low-quality reads that contained adapters, empty reads, or reads with unknown sequences “N” from the raw data. Quality control of reads was assessed using FastQC program (version 0.11.3) [23]. Subsequently, the clean reads from each cDNA library were aligned to the reference genome sequences of *R. pedestris* (https://www.ncbi.nlm.nih.gov/datasets/genome/GCA_019009955.1/ (accessed on 4 September 2022)) [24] using Hierarchical Indexing for Spliced Alignment of Transcripts (HISAT2) [25]. The low-quality alignments were filtered with Sequence Alignment/Map tools (SAMtools) [26].

### 2.4. Identification of Differentially Expressed Genes (DEGs) and Principal Component Analysis (PCA)

To investigate the transcriptomic changes in the gut under different food resources, transcripts per million (TPM) expression values were calculated using cufflink [27]. The DESeq2 (http://bioconductor.org/packages/release/bioc/vignettes/Glimma/inst/doc/DESeq2.html (accessed on 4 September 2022)) was used to identify the differentially expressed genes with default parameters. Based on these statistical analyses, genes meeting the criteria of *p*-value < 0.05 and an absolute value of the log_2_ ratio > 1 were deemed as DEGs. To reveal overall differences in gene expression patterns among different transcriptomes, R function plotPCA (github.com/franco-ye/TestRepository/blob/main/PCA_by_deseq2.R (accessed on 4 September 2022) was used to perform PCA.

### 2.5. Enrichment Analysis

The GO enrichment analyses were performed using TBtools software v1.0697 [28]. In this software, enriched *p*-values were calculated according to one-sided hypergeometric test: $P=1-\sum_{i=0}^{m-1}\left(\frac{\left(\begin{array}{c} M\\ i \end{array}\right)\left(\begin{array}{c} N-M\\ n-i \end{array}\right)\;}{\left(\begin{array}{c} N\\ n \end{array}\right)}\right)\;$, with *N* representing the number of genes with GO annotation, *n* representing the number of DEGs in *N*, *M* representing the number of genes in each GO term, and *m* representing the number of DEGs in each GO term. The significant pathways were defined based on a corrected *p*-value ≤ 0.05.

### 2.6. qRT-PCR Analysis

The gut samples of ten 5th instar nymphs were collected as described above. Total RNA was isolated with RNAiso plus. Three independent biological replicates with each repeated thrice were performed. To generate the first-strand cDNA and eliminate any genomic DNA contamination, the HiScript II Q RT SuperMix (Vazyme, Nanjing, China) was employed. Real-time quantitative PCR (qRT-PCR) was conducted on a Roche Light Cycler^®^ 480 Real-Time PCR System using the SYBR Green Supermix Kit (Yeasen, Shanghai, China). The qRT-PCR reaction program consisted of an initial denaturation step at 95 °C for 5 min, followed by 40 cycles of amplification at 95 °C for 10 s and 60 °C for 30 s. A relative quantitative method (2^−ΔΔCt^) was used to evaluate quantitative variation. The top ten DEGs associated with insect detoxification and digestion were selected, and the gene-specific primers used for qRT-PCR were designed using Primer 5.0 software (Appendix A). *R. pedestris* actin was used as an internal control. SPSS 22.0 was used to determine the differences in relative expression of DEGs among the three groups (one-way *t* test followed by Tukey’s multiple comparisons test, *p* < 0.05).

## 3. Results

### 3.1. Infestation of R. pedestris on Different Food Resources

The soybean leaves, pods, and beans infested by *R. pedestris* were investigated. The beans and pods became crinkled, while necrosis spots appeared on the leaves (Figure 1). SEM was employed to further investigate the feeding behaviors of *R. pedestris*. The results revealed the presence of numerous salivary sheaths on the surface of beans, pods, and leaves (Figure 1), indicating the successful feeding of *R. pedestris* on these soybean tissues. Notably, the appearance of salivary sheaths on different tissues exhibited variations. Specifically, the salivary sheaths on leaves were short and incomplete, whereas those on pods and beans were long and complete (Figure 1), suggesting that *R. pedestris* displays distinct feeding patterns and behaviors on different plant tissues. 

### 3.2. Overview of RNA Sequencing Data

To gain deeper insights into the adaptation of *R. pedestris* to different food resources, gut transcriptomic analyses of *R. pedestris* fed on beans, pods, and leaves were performed (hereafter, G—bean, G—pod, and G—leaf). A total of 19,006 genes were determined to be expressed in guts (Appendix A). PCA demonstrated that the three biological replicates from each treatment were well clustered, indicating the non-negligible influence of food resources on *R. pedestris*. In detail, the first principal component (PC1) captures the 37% contributing factor of variation, which well distinguished the G—bean from the G—leaf plus the G—pod. The second principal component (PC2) captures the 29% contributing factor of variation, and the G—leaf can be distinguished from the G—bean plus the G—pod in this case (Figure 2).

### 3.3. Analysis of Differently Expressed Genes (DEGs)

To identify candidate genes that contribute to the adaptation of *R. pedestris* to different food resources, gene expression patterns were analyzed in three comparison groups, including G—bean vs. G—leaf, G—pod vs. G—leaf, and G—pod vs. G—bean.

In the comparison between the G—pod and G—leaf, the expression of 443 genes was higher expressed in the G—leaf, while the expression of 355 genes was higher in the G—pods (Figure 3, Appendix A. Among the top 10 higher DEGs in the G—leaf, 4 were associated with serine protease (Appendix A). Other genes, such as vitellogenin, hexamerin, and trialysin, showed significant higher expression in the G—pod. Enrichment analysis revealed that the highly expressed genes in the G—pod were mainly associated with amino acid metabolism, carbohydrate metabolism, lipid metabolism, and peroxisome. On the other hand, the lower-expressed genes, such as cuticle protein, ribosomal protein, low-density lipoprotein receptor, ABC transporter, 4-coumarate-CoA ligase, and sodium-coupled monocarboxylate transporter, showed lower expression in the G—pod. Enrichment analysis demonstrated that the majority of highly expressed genes in the G—leaf were involved in ribosome biogenesis, mitochondrial biogenesis, and ion channels. These findings suggest that hydrolytic enzymes are more active in catalyzing proteins and lipids in soybean pods, while basal biosynthetic processes are elevated when feeding on soybean leaves (Figure 4, Appendix A).

In the comparison between the G—bean and G—leaf, a total of 690 DEGs were identified, including 366 higher—expressed genes in the G—bean and 324 higher—expressed genes in the G—leaf. Similar to the comparison of the G—pod and G—leaf, gene expressions such as vitellogenin, hexamerin, venom serine protease, peroxidase, and carboxypeptidase were significantly higher in the G—bean, while expressions of cuticle protein, synaptic vesicle protein, ribosomal protein, solute carrier organic anion transporter, and ABC transporter were significantly lower in the G—bean (Appendix A). Enrichment analysis yielded similar results, with the majority of highly expressed genes in the G—bean involved in amino acid metabolism, carbohydrate metabolism, lipid metabolism, transporter activity, and peroxisome function. In contrast, the majority of highly expressed genes in the G—leaf were associated with ribosome biogenesis, proteasome activity, translation, and mitochondrial biogenesis (Appendix A).

Additionally, we compared the gene expressions between the G—pod and G—bean. The results revealed that the expression levels of 271 genes were significantly decreased, while 277 genes were higher—expressed when feeding on soybean pods (Figure 3, Appendix A). The higher—expressed genes were primarily involved in protein binding, anatomical structure development, and developmental processes (Appendix A). On the other hand, the majority of lower-expressed genes were associated with hydrolase activity, extracellular region, and catabolic processes (Figure 4, Appendix A). These findings suggest that the components involved in development are induced when *R. pedestris* feed on soybean pods, indicating that the fitness of pods may be higher compared to that of beans.

### 3.4. Expression Profiles of DEGs Accsociated with Detoxificatiion and Digestion

The top ten DEGs associated with insect detoxification and digestion were selected for qRT-PCR determination, including apolipoprotein D, cytochrome P450 6a2-like, sonic hedgehog protein, vitellogenin, an unknown secreted protein, enhancer of split mbeta protein-like, mucin-17 like protein, carcinine transporter, and *cathepsin* L.

Three genes (apolipoprotein D, presequence protease, and sonic hedgehog protein) (Figure 5A,B,D) showed high expression levels in the G—pod and G—bean when compared with that of the G—leaf. Cytochrome P450 6a2 like only displayed the highest expression when *R. pedestris* were feeding on beans, with no significant difference observed when they were feeding on leaves and pods (Figure 5C). The expression of vitellogenin was higher in the G—bean than the G—pod (Figure 5E). The expression levels of cathepsin L, enhancer of split mbeta protein-like, mucin-17 like protein, carcinine transporter were significantly higher when *R. pedestris* fed on soybean leaves compared to the other groups (Figure 5G–J). Overall, the qRT-PCR data for eight out of the ten genes were consistent with the RNA-seq data, indicating the reliability of RNA-seq. However, there are inconsistencies in the expression levels of cytochrome P450 6a2-like and a secreted protein between the two methods (Figure 5C,F). It is possible that these differences are caused by using different samples in the two methods, leading to variations in gene expression results.

## 4. Discussion

Understanding the mechanisms underlying the adaptation of *R. pedestris* to different plant tissues is crucial for gaining insights into their behavior in the field and developing strategies to mitigate crop damage caused by these pests. In this study, we analyzed the gene expression patterns in the guts of *R. pedestris* that feed on soybean leaves, pods, and beans. Our findings highlight the significance of genes involved in catalytic activity, hydrolase activity, basal biosynthetic processes, and detoxification in the insect adaptation to these diverse food sources.

*R. pedestris* can successfully feed and probe on different soybean tissues, with abundant salivary sheath left on the surface of the food. However, the appearance of the salivary sheath varied on pods, beans, and soybean leaves. The palatability is closely related to the chemical composition of these food resources [29,30]. The gel saliva, which is the main constitute of salivary sheath, is continuously secreted along with insect feeding. The short salivary sheath indicates a shorter probing period during *R. pedestris* feeding [31]. The salivary sheath on leaves was shorter and incomplete, suggesting lower palatability compared to pods and beans.

Herein, we found enhanced expression of genes associated with catalytic activity, hydrolase activity, and peptidase activity when *R. pedestris* fed on beans and pods. Both tissues are rich in amino acids and soluble proteins [32,33], and the activation of these enzymes ensures normal metabolism and an adequate energy supply. The most abundant genes identified were cathepsin L, venom serine carboxypeptidase, and cathepsin F (Appendix A), which are known to play a direct role in insect digestion [34,35]. Additionally, alpha-glucosidase 2, which is involved in digestion, showed high abundance in the gut of *R. pedestris* feeding on beans and pods [36]. Furthermore, vitellogenins, which are precursors of the major egg storage protein vitellin [37,38], were highly expressed when the insects fed on beans and pods. The enhanced expression of these genes while *R. pedestris* was feeding on pods and beans indicates that the insects thrive better on beans and pods compared to soybean leaves, which is consistent with previous conclusions [32,33].

On the other hand, components related to cellular processes, localization, and transport showed higher expression when *R. pedestris* fed on soybean leaves, including ABC transporters and uricase. ABC transporters play critical roles in insect development and immunity [39], and they have been shown to assist herbivorous insects in adapting to different host plants by promoting the degradation of plant secondary metabolites [40,41,42]. The increased expression of ABC transporter may promote the degradation of *R. pedestris* to secondary metabolites in soybean leaves. Moreover, several genes related to lipid metabolism, such as 4-coumarate-CoA ligase 1-like and RNA-binding protein NOB1, exhibited increased expression when *R. pedestris* fed on soybean leaves. These genes are involved in the breakdown of lipids, which are defensive metabolites against herbivores [43,44]. The increased expression of these genes enables *R. pedestris* to successfully feed on soybean leaves by overcoming the chemical defense mechanisms.

Previous studies have provided valuable references for exploring the adaptation mechanisms of herbivorous insects to their host plants. For example, Yun et al. [45] compared the metabolomics of soybean leaves and beans, and they found that beans had higher levels of amino acids and soluble proteins, while leaves had higher levels of flavonoids and phenolic acids. This is consistent with our results, which showed that the expression of genes related to digestion and growth was expressed significantly higher when *R. pedestris* fed on beans and pods. Similar findings have been reported in other insect species, such as the sugarcane borer (*Diatraea saccharalis*), *Chilo suppressalis*, and *Trichoplusia ni*, in response to different food resources [46,47,48]. It has been observed that the longevity of *R. pedestris* fed on soybean leaves is significantly shorter than that of bugs fed on pods and beans, which is positively correlated with the concentration of amino acids and proteins [32,33]. Additionally, several genes associated with growth, such as apolipoprotein D and Sonic hedgehog protein [49,50,51], showed significant higher expression when *R. pedestris* fed on pods and beans. These results support the previous conclusion that pods and beans are more favorable food sources for *R. pedestris* compared to soybean leaves [32,33].

By comparing the transcriptomics of the insect gut when fed on different resources, we identified several key metabolic pathways that favor bean bug feeding on soybean leaves, pods, and beans. This enables *R. pedestris* to extend their occurrence period, leading to population growth in the field. Furthermore, we identified several candidate genes that potentially enhance the feeding behaviors of *R. pedestris* on leaves. These digestive genes can be employed as potential targets to minimize the population of *R. pedestris* in the field, thereby reducing the damage caused by these pests and preventing outbreaks of infestations.

## Figures and Tables

**Figure 1 insects-14-00739-f001:**
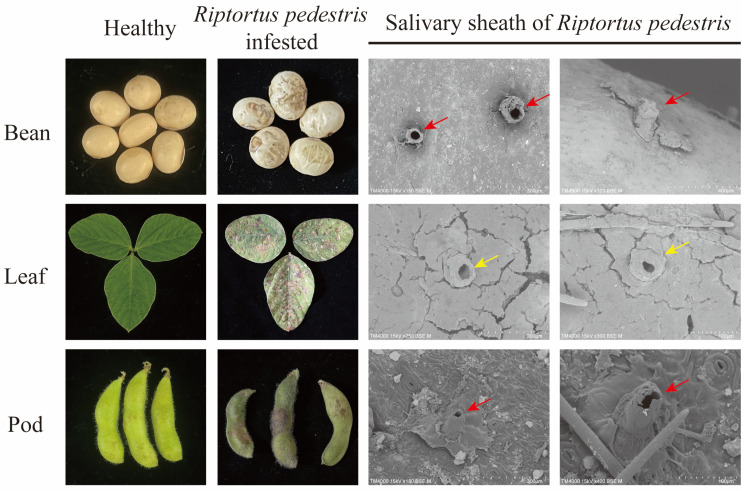
Soybean leaves, pods, and beans infested by *Riptortus pedestris*. Bean bug feeding induces necrosis spots on leaves and extracts nutrients from pods and beans, resulting in a significant decrease in soybean production. The healthy bean, leaf, and pod without *R. pedestris* infestation were used as controls. The morphology of each sample was photographed using a Canon EOS 80D camera. Salivary sheath of the bean bug was observed using scanning electron microscopy (SEMTM4000 II plus, Hitachi). The salivary sheath left on leaf was short and incomplete (indicated by yellow arrows), whereas those on pods and beans were long and complete (indicated by red arrows). For each tissue, more than five biological replicates were performed, and similar results were found. Two representative images for each tissue were displayed.

**Figure 2 insects-14-00739-f002:**
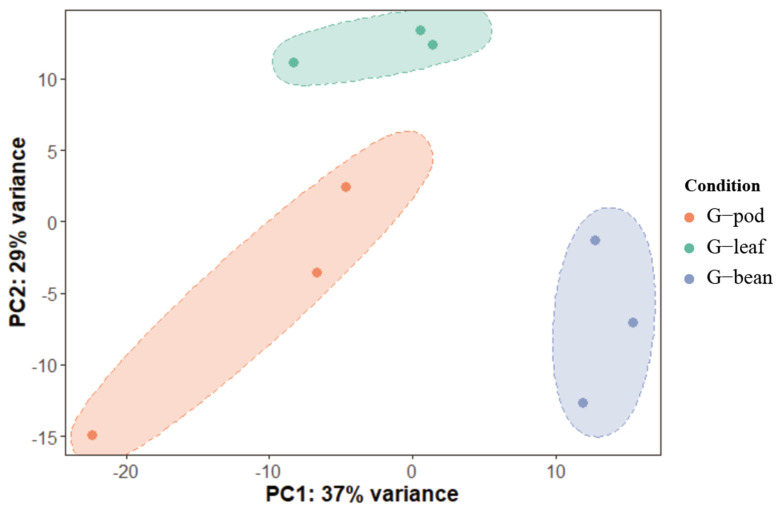
Principal components analysis of gene expression patterns in *Riptortus pedestris* feeding on different hosts. G—pod, G—leaf, and G—bean represent gut collections of bean bugs that fed on pods, leaves, and beans, respectively. The first two principal components (PC1 and PC2) based on transcriptomic results are shown.

**Figure 3 insects-14-00739-f003:**
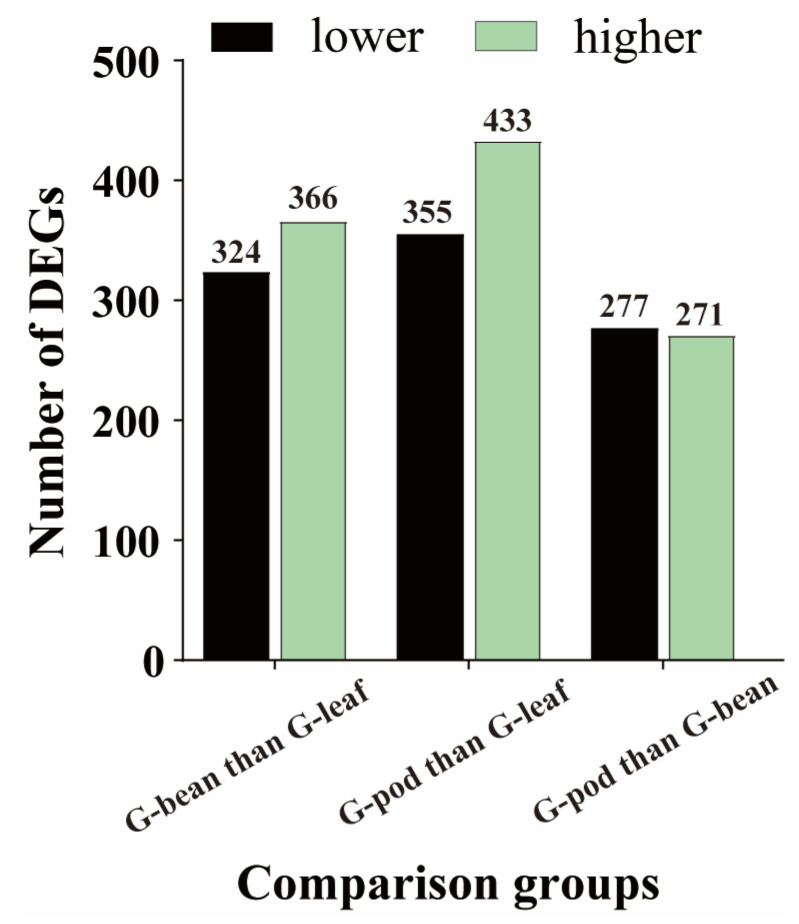
Statistics of differentially expressed genes (DEGs) in different comparisons. The genes with a fold change > 2 and *p*-value < 0.05 were included. The black columns represent the number of genes lower expressed in each group, while the green columns represent the number of genes higher expressed in each comparison.

**Figure 4 insects-14-00739-f004:**
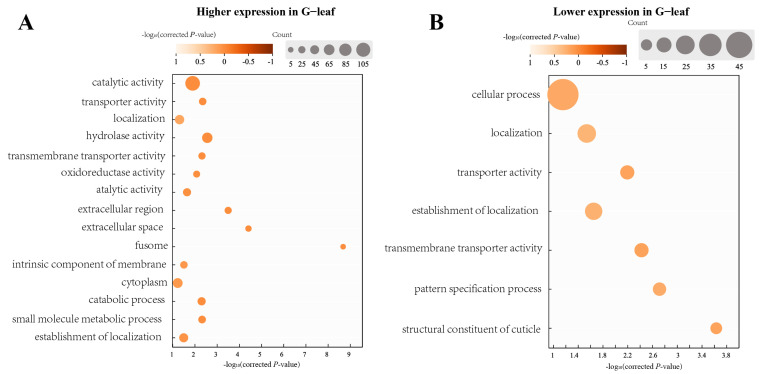
Enrichment analysis of DEGs in different comparisons. (**A**) DEGs higher expressed in G—leaf than these of G—bean/G—pod. (**B**) DEGs lower expressed in G—leaf than these of G—bean/G—pod. Corrected *p*-values of each term were calculated according to one-sided hypergeometric test using TBtools software. The size of circles indicates the number of DEGs. The color reflects the −log_10_ (corrected *p* value). Details in GO enrichment analyses were displayed in Appendix A.

**Figure 5 insects-14-00739-f005:**
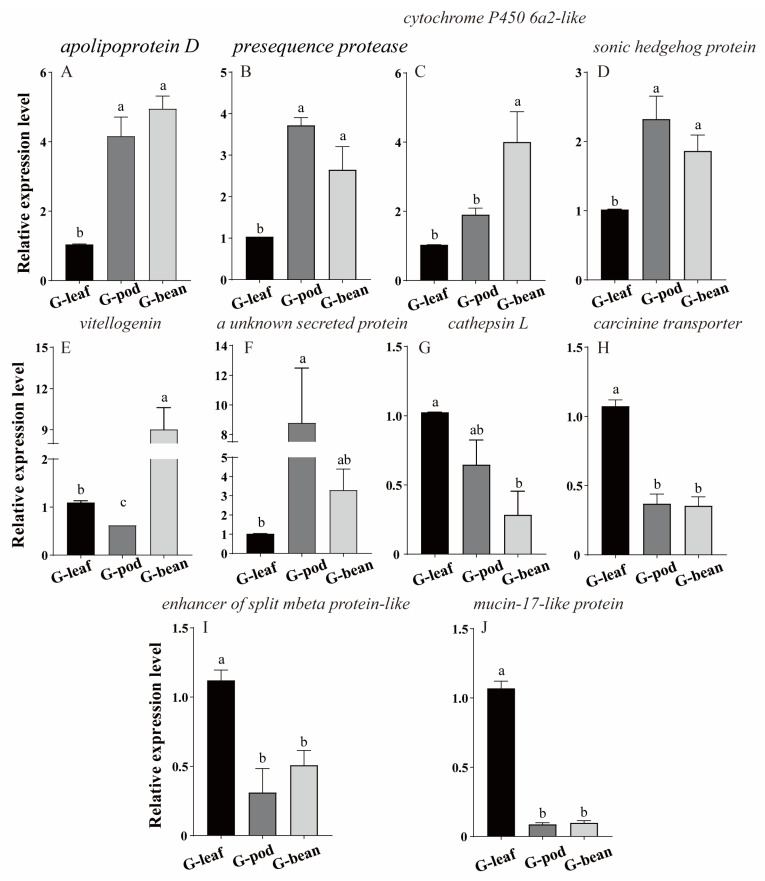
Gene expression quantified by qRT-PCR. Differences were calculated using one-way ANOVA test followed by Tukey’s multiple comparisons test. Different letters indicate significant differences between comparisons (*p* < 0.05). Gene names are marked above each diagram. (**A**–**J**) show the relative expression level of ten genes in different groups respectively. Gene names are displayed above the subfigures.

## Data Availability

All sequencing data generated in this study were submitted to the National Genomics Data Center under accession number: PRJCA019207.

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
