# Peer review of "Comparative Transcriptomic Analysis Reveals Adaptation Mechanisms of Bean Bug Riptortus pedestris to Different Food Resources"

_insects, 2023, doi:10.3390/insects14090739_

Round 1

Reviewer 1 Report

The authors investigated the transcriptomic effects of the bean bug, Riptortus pedestris, when feeding on the three soya bean tissues: bean, pod and leaf. They found several differences in gut gene expression between the three feeding regimes.

The manuscript is overall well written and clearly argued. The results are quite conclusive and well interpreted. I think it merits publication in this journal but requires some revisions first.

-Section 2.3: Please provide more details on the sampling procedure. You write that forty nymphs were fed with the three tissues. But how many on each? And how does this translate to only 3 biological replicates per tissue? Were the individuals pooled? If so please provide details. A table could help.

-Section 2.4 please provide a citation for DEGseq. Is it legitimate to use TPM for this programme? DESeq2, for example, requires raw counts.

-Section 2.5 - this requires more information on the methods. How was the enrichment calculated? With which package or statistical method, were p-values corrected? etc.

-Section 2.5 - how many replicates were used for the qRT-PCR? Pooled individuals?

-Section 3.2 - the PCs are incorrectly interpreted. PC1 distinguishes bean from leaf+pod; PC2 leaf from bean+pod. Correct "leave" with "leaf" in the figure

-Section 3.3 - here and elsewhere in the manuscript, what is named as up- or down-regulated is arbitrary and often determined by the differential expression package. In fact, it is not clear if there is an up- or down-regulation, just that the expression is higher in one condition and simultaneously lower in the other. Using up- and down- here and in figures 3+4 makes it difficult to interpret in which tissue expression is higher. I would suggest to explicitly state significantly higher in leaf than in bean, for example. In figure 3, you could label each bar with the corresponding tissue in which expression is higher - otherwise it is impossible to understand what up and down mean.

-Section 3.3 - More details are needed for the enrichment results in this section. Is this based on GOterms and/or KEGG terms - provide tables of full results. There are some in the suppmat - these should be referenced here.

- Figure 4 - this figure is very confusing. What does down- and upregulated mean? In comparison to what? And how can a downregulated gene have a positive or a negative relative expression value (scale from 3 to -3)? And vice versa for upregulated. Also, it is not clear how relative expression has been calculated. If these are just all the significant DEGs from the three comparisons, then they belong in one single heatmap, because, as I explain before, the classification into up- and down-regulation is arbitrary. How are the functional categories determined? With more details in the caption and methods and putting all into one heatmap, this could be a valuable figure. Please remove the word "analysis" from the caption as heatmap is a method of illustration not an analysis.

-Section 3.4 - how were the genes selected for qRT-PCR? There were hundreds of DEGs, so please be more explicit as to how these were selected. Also, please tone down the need to validate RNAseq. It is a well-established method which is more rigorous than qRT-PCR, since stats are run across all genes. It's nice that some interesting genes were additionally tested with confirming results but maybe leave out "indicating the reliability of RNAseq".

Well written with good English. Just a few typos that can be easily corrected.

Author Response

#Review 1

The authors investigated the transcriptomic effects of the bean bug, Riptortus pedestris, when feeding on the three soya bean tissues: bean, pod and leaf. They found several differences in gut gene expression between the three feeding regimes.

The manuscript is overall well written and clearly argued. The results are quite conclusive and well interpreted. I think it merits publication in this journal but requires some revisions first.

Response: We greatly appreciate your valuable comments and constructive suggestion to our work.

-Section 2.3: Please provide more details on the sampling procedure. You write that forty nymphs were fed with the three tissues. But how many on each? And how does this translate to only 3 biological replicates per tissue? Were the individuals pooled? If so please provide details. A table could help.

Response: Done. Thanks for your suggestion, we have added descriptions about the replications as follow: “A group of forty 5th instar nymphs of R. pedestris were fed with pods, beans, and leaves for a duration of four days in separate net cages(25cm×25cm×25cm). The insect guts were dissected with a pair of forceps, and were transferred into RNAiso plus (TaKaRa, Dalian, China). Guts dissected from ten insects were pooled and regard as one replicate”.

-Section 2.4 please provide a citation for DEGseq. Is it legitimate to use TPM for this programme? DESeq2, for example, requires raw counts.

Response: Done. Thank you for pointing out this. In the revised version, we provide the hyperlink of DESeq2 in the method section, and describe the method of generating TPM matrix as follow: “To investigate the transcriptomic changes in the gut under different food resources, transcripts per million (TPM) expression values were calculated using cufflink [26]. The DESeq2 (http://bioconductor.org/packages/release/bioc/vignettes/Glimma/inst/doc/DESeq2.html) was used to identify the differentially expressed genes with default parameters. Based on these statistical analyses, genes meeting the criteria of P-value < 0.05 and an absolute value of the log2 ratio>1 were deemed as DEGs”.

-Section 2.5 - this requires more information on the methods. How was the enrichment calculated? With which package or statistical method, were p-values corrected? etc.

Response: Done. Thank you for pointing out this. In the revised version, we provide the details in enrichment analyses as follow: “The GO enrichment analyses were performed using TBtools software v1.0697[27]. In this software, enriched P-values were calculated according to one-sided hypergeometric test: , with N represents the number of gene with GO annotation, n represents the number of DEGs in N, M represents the number of genes in each GO term, m represents the number of DEGs in each GO term. The significant pathways were defined based on a corrected P-value ≤ 0.05”.

-Section 2.5 - how many replicates were used for the qRT-PCR? Pooled individuals?

Response: Done. In the revised version, we have added details about the replicates in qRT-PCR as follow: “Three independent biological replicates with each repeated thrice were performed”.

-Section 3.2 - the PCs are incorrectly interpreted. PC1 distinguishes bean from leaf+pod; PC2 leaf from bean+pod. Correct "leave" with "leaf" in the figure

Response: Done. Sorry about this inaccurate interpretation. We corrected it in the revised version.

-Section 3.3 - here and elsewhere in the manuscript, what is named as up- or down-regulated is arbitrary and often determined by the differential expression package. In fact, it is not clear if there is an up- or down-regulation, just that the expression is higher in one condition and simultaneously lower in the other. Using up- and down- here and in figures 3+4 makes it difficult to interpret in which tissue expression is higher. I would suggest to explicitly state significantly higher in leaf than in bean, for example. In figure 3, you could label each bar with the corresponding tissue in which expression is higher - otherwise it is impossible to understand what up and down mean.

Response: Done. Thanks for your suggestion. In the revised version, we modified those labels in figures, and check the description of up- or down-regulation throughout the manuscript.

-Section 3.3 - More details are needed for the enrichment results in this section. Is this based on GO terms and/or KEGG terms - provide tables of full results. There are some in the suppmat - these should be referenced here.

Response: Done. Thanks for your suggestion, we have added citation in this section. Some results were presented in the supplementary files.

- Figure 4 - this figure is very confusing. What does down- and upregulated mean? In comparison to what? And how can a downregulated gene have a positive or a negative relative expression value (scale from 3 to -3)? And vice versa for upregulated. Also, it is not clear how relative expression has been calculated. If these are just all the significant DEGs from the three comparisons, then they belong in one single heatmap, because, as I explain before, the classification into up- and down-regulation is arbitrary. How are the functional categories determined? With more details in the caption and methods and putting all into one heatmap, this could be a valuable figure. Please remove the word "analysis" from the caption as heatmap is a method of illustration not an analysis.

Response: Done: Thanks for your suggestion, we have added descriptions about the GO enrichment analyses in the M&M and Results sections. Additionally, figures were modified to clearly present the expression of genes in different samples.

-Section 3.4 - how were the genes selected for qRT-PCR? There were hundreds of DEGs, so please be more explicit as to how these were selected. Also, please tone down the need to validate RNAseq. It is a well-established method which is more rigorous than qRT-PCR, since stats are run across all genes. It's nice that some interesting genes were additionally tested with confirming results but maybe leave out "indicating the reliability of RNAseq".

Response: Done. Thanks for your suggestion. In the revised version, we added the criteria in selecting genes for qRT-PCR. Also, we remove the statement of “indicating the reliability of RNAseq”

Comments on the Quality of English Language

Well written with good English. Just a few typos that can be easily corrected.

Response: Thank you for reviewing our manuscript. We carefully check the English language in the revised version.

Reviewer 2 Report

Dear Author/s

Submitted manuscript Insects-2497244-v1 authored by Zhang et al has done nice work and meticulously written the manuscript. The Authors have investigated an interesting topic related to insect physiology/toxicology and possible health risks, and the theme has been properly described. I would like to congratulate authors for the good quality of the article, the literature reported used to write the paper, and for the clear and appropriate structure. The manuscript is well written, presented and discussed, and understandable to a specialist readership.

I am pointing out some minor corrections here, either need to be answered or amended in the current draft.

1.     In method section, how did raw counts were further converted/analyzed to the fragments per kilobase per million mapped reads (FPKM) module is missing.

2.     The obtained FPKM values were log10-transformed (by adding 1 to it) before generating a heatmap on rows is also not clear.

3.     Differential expression analysis was performed for selected groups using which method? I would suggest that take the help/cite this paper https://doi.org/10.3390/pathogens10091127

4.     Consecutively, Figure 4 legend and methodology needs little more description.

5.     I would recommend that figure 5 should be further categorized in A,B and C and mention these gene expression profile in the result accordingly. Improvise the quality of figures.

6.     Figure 2 is the quality control of your RNA-seq data, you can add more parameters or analytical attributes in it. Take the help/cite of above-mentioned paper.

7.     There is no mention of the supplementary file S3-4 in the main text.

Make sure the language of the paper is appropriate. Validate other mistakes at your end and submit the clearer and more readable version.

All the best.

Make sure the language of the paper is appropriate. Validate other mistakes at your end and submit the clearer and more readable version.

Author Response

#Review 2

Submitted manuscript Insects-2497244-v1 authored by Zhang et al has done nice work and meticulously written the manuscript. The Authors have investigated an interesting topic related to insect physiology/toxicology and possible health risks, and the theme has been properly described. I would like to congratulate authors for the good quality of the article, the literature reported used to write the paper, and for the clear and appropriate structure. The manuscript is well written, presented and discussed, and understandable to a specialist readership.

Response: We greatly appreciate your valuable comments and constructive suggestion to our work.

I am pointing out some minor corrections here, either need to be answered or amended in the current draft.

  1. In method section, how did raw counts were further converted/analyzed to the fragments per kilobase per million mapped reads (FPKM) module is missing.

Response: Done. Thanks for pointing out this. In this study, we use the software Cufflink (Trapnell et al., Nat Protocol, 2012) to convert raw count data to transcripts per million (TPM). In the revised version, we describe this procedure, and cite the corresponding references.

  1. The obtained FPKM values were log10-transformed (by adding 1 to it) before generating a heatmap on rows is also not clear.

Response: Thank you for pointing out this. In analysis of TPM data, there is no log10-transformed procedure. In the revised version, we describe the methods in generating the TPM and DEGs as follow: “To investigate the transcriptomic changes in the gut under different food re-sources, transcripts per million (TPM) expression values were calculated using cufflink [26]. The DESeq2 (http://bioconductor.org/packages/release/bioc/vignettes/Glimma/inst/doc/DESeq2.html) was used to identify the differentially expressed genes with default parameters. Based on these statistical analyses, genes meeting the criteria of P-value < 0.05 and an absolute value of the log2 ratio>1 were deemed as DEGs”.

  1. Differential expression analysis was performed for selected groups using which method? I would suggest that take the help/cite this paper https://doi.org/10.3390/pathogens10091127

Response: Thank you for your suggestion. Given the differences in performing DEG analysis, we did not cite this work in DEG analysis section. However, this work reveals the importance of ABC transporters in insect development and immunity, which might be also associated with insect adapting on different hosts. Therefore, we cite this work in the discussion section.

  1. Consecutively, Figure 4 legend and methodology needs little more description.

Response: Done. Thanks for your suggestion, the GO enrichment analysis were performed with TBtools software. The categories of DEGs were labeled based on the enrichment analysis results. We have added descriptions in Results and M&M sections.

  1. I would recommend that figure 5 should be further categorized in A,B and C and mention these gene expression profile in the result accordingly. Improvise the quality of figures.

Response: Done. In the revised version, the figure 5 has been modified according to your suggestion. Also, we added citation of A, B, C in the result section, when refers to these genes.

  1. Figure 2 is the quality control of your RNA-seq data, you can add more parameters or analytical attributes in it. Take the help/cite of above-mentioned paper.

Response: Done. Thanks for your suggestion. In the revised version, we provide the detail about the methods used in performing PCA, and provide the code hyperlink in the M&M section.

  1. There is no mention of the supplementary file S3-4 in the main text.

Response: Done. Thanks for pointing out this. In the revised version, we added citation in the manuscript.

Make sure the language of the paper is appropriate. Validate other mistakes at your end and submit the clearer and more readable version.

All the best.

Response: Thank you for reviewing our manuscript. We carefully check the English language in the revised version.

Reviewer 3 Report

This study presents a transcriptomic analysis of the gut of the bean bug, Riptortus pedestris, fed on different parts of the soybean plant (leaves, pods, and beans). The authors carried out mRNA sequencing from the samples and performed pairwise comparisons for differential gene expression analysis. Notably, the study highlights increased expression of genes related to digestive enzymes like cathepsins, serine proteases, and lipases in bugs fed on pods and beans.  Conversely, upregulation of detoxification enzymes like ABC transporters and 4-coumarate-CoA ligase was observed in bugs fed on leaves.

Broad comments:

The introduction requires an expansion to elucidate the rationale behind the study. While the authors mentioned the economic importance of the bean bug because of its damage to soybean production. They also cited different interesting papers and mentioned briefly the comparative metabolomic study between different parts of soybean plants and the effect on the growth and development of the bean bug. These are excellent points, however, authors need to expand on them to illustrate why it is important to understand the mechanisms behind feeding on the different plant parts, what is the biological relevance and how can we use this information. For example, as it is mentioned that the bean bug is a very problematic pest, can we use this information to strategize better control strategies? It is important to mention the clear rationale in the introduction.

The materials and method section is well structured; however, details are not adequately presented, please write in detail on every step of sample collection to analysis so that readers, if want, can easily replicate your study.

The data availability section is missing from the manuscript. It’s important the sequencing data are stored in a public repository and made publicly available or if in case this cannot be done, explain the reason.

Specific comments

Line 71-76: Please provide the coordinates of where R. pedistris were originally collected and also the scientific name of the host plant on which it was reared.

Line 77-84: Provide more details on how this is done. How did the authors feed the insects with different plant parts? Did they keep the insects in separate containers (plastic?, what size containers?), and how much of plant parts were provided? what are stubs and how were the plant parts attached? Is it air drying or some other way of drying in line 82? Explain the process of sample preparation for SEM or cite protocols you followed.

Line 85-91: Same comment as above, how were the insects fed? did the authors keep all the insects for one treatment in one container while feeding? Write about the container, size, and how was the aeration in the container maintained. Or it was on an insect-rearing cage with nets?

Did the authors dissect insects under the dissection microscope? If yes, which one? Were any buffers (like PBS?) used during dissection and/or kept in cold? Was the gut cleaned for the inside debris and insect hemolymph and tissues with any buffers, if not, why? Did the authors use the whole gut or a specific region of the gut (foregut, midgut, hindgut) mention it? What is the RNA-extraction protocol used, provide the name and catalog number of the kit/reagents, Were there any DNAse steps involved? How did the authors measure the RNA integrity and quantity? was any threshold used for the minimum quality of RNA to use?

Line 98: Provide the PCR conditions, how many cycles?

Line 100-103: how did the authors filter the poor-quality reads, what was the threshold set to determine if the reads were poor quality? What is the internal software? The reference cited here (number 21) is not about the software, correct it.  Write down the version of HISAT used.

Line 104-108: Provide more details on how you performed the DEG analysis.

Eg. Mention any types of normalization done. Did the authors filter out the genes that were low/sparsely expressed?

Line 109-115: Provide detailed methodologies. How did the authors carry out the enrichment analysis? What software was used? What type of analysis was carried out, and what were the background genes for the enrichment analysis?

Line 113 -115 is unnecessary in this section.

Line 116- 127: How did the authors normalize the expression data, write about any housekeeping genes used. How many biological and technical replicates were used?

Line 129 – 138: Point out the long and short salivary sheaths in the figure (with arrows) what is the incomplete sheath on leaves and the complete on pods and beans, please elaborate.

Figure 1: The healthy and infested samples were incubated both for 4 days in similar conditions except for the presence of insects on the infested samples. The two columns of the SEM are they both are from infested samples? Do you have SEM from healthy samples to compare with?

Line 141-146: Italicize scientific names. Explain the difference between the two columns of the SEM image. Are these SEM images representative of many taken, how many? Give the model of the SEM microscope and the other camera used in imaging in the figure legends.

Line 156 – 161: G-pod cluster and G-leaf cluster are not separated on PC1 similarly, G-bean and G-pod are not separated on PC2, correct the sentences here.

Line 164: did the authors mean to write G-leaves or G-leaf instead of G-leave? same in the figure legend.

Line 167- 213

Plot the enrichment analysis results in more meaningful and easier-to-visualize figures showing the enrichment scores and number of genes involved. Figure 4 does not show the enrichment scores. The authors can use heatmaps with a clustering approach to show the expression patterns of the genes in different treatment categories. Tables in the supplementary materials don’t make the picture clear as the authors are presenting it in the text.

Line 215-218: Write it in the method section. How did the authors select these genes? Randomly?

Line 227- 231: What was the number of biological replicates for the RT-PCR? Is the inconsistency in the expression due to not enough biological and technical replicates?

Line 234-236: The post hoc test used?

Line 248-249: provide a reference for this line: “The palatability is closely related to the chemical composition of these food resources”.

Line 249-250: How do the authors relate the shorter salivary sheath with the chemical composition and palatability? Is something missing here? Explain what incomplete and complete salivary sheath is.

Line 260: Can the authors elaborate, on what they mean by positive change? Do they mean these genes are upregulated while feeding on pods and beans but not while feeding on leaves?

Line 317: Italicize the scientific name Helicoverpa armigera. Check the entire references for similar mistakes. Eg: Line 326, 328, 330. Check for inconsistencies on the formatting of the references.

Writing is understandable. Find my comments above.

Author Response

#Review 3

This study presents a transcriptomic analysis of the gut of the bean bug, Riptortus pedestris, fed on different parts of the soybean plant (leaves, pods, and beans). The authors carried out mRNA sequencing from the samples and performed pairwise comparisons for differential gene expression analysis. Notably, the study highlights increased expression of genes related to digestive enzymes like cathepsins, serine proteases, and lipases in bugs fed on pods and beans.  Conversely, upregulation of detoxification enzymes like ABC transporters and 4-coumarate-CoA ligase was observed in bugs fed on leaves.

Response: We greatly appreciate your valuable comments and constructive suggestion to our work.

Broad comments:

The introduction requires an expansion to elucidate the rationale behind the study. While the authors mentioned the economic importance of the bean bug because of its damage to soybean production. They also cited different interesting papers and mentioned briefly the comparative metabolomic study between different parts of soybean plants and the effect on the growth and development of the bean bug. These are excellent points, however, authors need to expand on them to illustrate why it is important to understand the mechanisms behind feeding on the different plant parts, what is the biological relevance and how can we use this information. For example, as it is mentioned that the bean bug is a very problematic pest, can we use this information to strategize better control strategies? It is important to mention the clear rationale in the introduction.

Response: Done. Thanks for your suggestion. In the revised version, we rewrote the introduction section as follow:

The bean bug, Riptortus pedestris, poses a substantial threat to soybean cultivation in Asia. Its remarkable ability to consume diverse soybean plant components—ranging from leaves to pods and beans—has significant implications for both crop quantity and quality [12-14]. The occurrence of the soybean Zhengqing, causing significant losses in soybean production, has been attributed to R. pedestris [15, 16]. In the wild, R. pedestris exhibits an intriguing dietary behavior. Its presence in fields precedes the maturity of soybean beans, necessitating a period of sustenance from the tender leaves of nascent soybean seedlings [16]. This dietary adaptation underscores the dynamic variance in phytochemical compositions across distinct soybean tissues. Metabolomic analyses have elucidated the chemical tapestry of soybean leaves, pods, and beans. Leaves are enriched with compounds such as flavonoids, phenolic acids, and alkaloids, which function as natural defense mechanisms against insect predation [17, 18]. In contrast, beans are reservoirs of lipids and proteins, offering a nutritionally distinct profile [17, 18]. Rahman and Lim found that R. pedestris exhibited reduced longevity and higher mortality when fed with pods and beans compared to beans alone [19]. Despite this knowledge, the mechanisms by which R. pedestris adapt to different hosts remain un-clear.

Understanding the intricate mechanisms that govern the feeding preferences of R. pedestris across various plant parts holds profound significance. This comprehension not only unravels the ecological intricacies of pest-host interactions but also offers a gateway to pragmatic agricultural solutions. For example, our recent study revealed key lipases linked to feeding and reproduction in R. pedestris [19]. These findings could guide us in using RNAi to control this harmful pest more effectively. As R. pedestris adaptation to varied diets is intricate, it is plausible that beyond lipases, other genes play pivotal roles in this process. Therefore, a comprehensive exploration of gene expression patterns using omics studies becomes imperative in unearthing the underlying mechanisms.

The materials and method section is well structured; however, details are not adequately presented, please write in detail on every step of sample collection to analysis so that readers, if want, can easily replicate your study.

Response: Done. Thanks for your suggestion. In the revised version, we provide more details in sample collection, DEG analysis, PCA analysis, enrichment analysis, and qPCR analysis. All readers would easily replicate our work based on our description.

The data availability section is missing from the manuscript. It’s important the sequencing data are stored in a public repository and made publicly available or if in case this cannot be done, explain the reason.

 Response: Done. In the revised version, we upload all data generated in this study to the National Genomics Data Center under accession number: PRJCA019207. We descript this in M&M section and Data Availability Statement.

Specific comments

Line 71-76: Please provide the coordinates of where R. pedistris were originally collected and also the scientific name of the host plant on which it was reared.

Response: Done.

Line 77-84: Provide more details on how this is done. How did the authors feed the insects with different plant parts? Did they keep the insects in separate containers (plastic?, what size containers?), and how much of plant parts were provided? what are stubs and how were the plant parts attached? Is it air drying or some other way of drying in line 82? Explain the process of sample preparation for SEM or cite protocols you followed.

Response: Done. Thanks for your suggestion. In the revised version, we added descriptions in this section. The stubs here mean a support structure of Scanning electron microscope. We cited corresponding reference.

Line 85-91: Same comment as above, how were the insects fed? did the authors keep all the insects for one treatment in one container while feeding? Write about the container, size, and how was the aeration in the container maintained. Or it was on an insect-rearing cage with nets?

Response: Done. Thanks for your suggestion. The insects were reared in net cages. In the revised version, we added statement about the method on insect rearing.

Did the authors dissect insects under the dissection microscope? If yes, which one? Were any buffers (like PBS?) used during dissection and/or kept in cold? Was the gut cleaned for the inside debris and insect hemolymph and tissues with any buffers, if not, why? Did the authors use the whole gut or a specific region of the gut (foregut, midgut, hindgut) mention it? What is the RNA-extraction protocol used, provide the name and catalog number of the kit/reagents, Were there any DNAse steps involved? How did the authors measure the RNA integrity and quantity? was any threshold used for the minimum quality of RNA to use?

Response: Done. Thanks for your suggestion. In the revised version, we detailed these procedure as follow: “A group of forty 5th instar nymphs of R. pedestris were fed with pods, beans, and leaves for a duration of four days in separate net cages (25cm×25cm×25cm). The insect guts were dissected in a phosphate-buffered saline (PBS) solution (137mM NaCl, 2.68mM KCl, 8.1mM Na2HPO4 and 1.47m MKH2PO4 at pH 7.4) with a pair of forceps under a stereomicroscope (Olympus SZX16; Tokyo, Japan). The dissected guts were washed in a clean PBS solution to avoid the contamination of other tissues. As it is difficult to remove the ingested food inside the lumen, we did not clean the gut lumen. Whole gut, including foregut, midgut, and hindgut was collected for analysis. The collected samples were transferred into RNAiso plus (TaKaRa, Dalian, China) and kept on ice during the dissection period. Guts dissected from ten insects were pooled and regard as one replicate. RNA was extracted according to manufactures’ protocol. The chloroform (Sango Biotechnology, Shanghai, China), isopropanol (Sango Biotechnolo-gy), and 75% ethanol (Sango Biotechnology) were used for RNA extraction. RNA integrity and quantity was assessed using the RNA Nano 6000 Assay Kit of the Bioanalyzer 2100 system (Agilent Technologies, CA, USA). The RNA samples were sent to Novogene (Tianjin, China) for library construction and Illumina sequencing.”

Line 98: Provide the PCR conditions, how many cycles?

Response: Done. We addressed that 16-cycle of PCR amplification was performed.

Line 100-103: how did the authors filter the poor-quality reads, what was the threshold set to determine if the reads were poor quality? What is the internal software? The reference cited here (number 21) is not about the software, correct it.  Write down the version of HISAT used.

Response: Done. In the revised version, we mentioned that “The internal software Fastp version 0.23.1 [22] was used to obtain clean reads by removing low-quality reads that contained adapters, empty reads or reads with un-known sequences “N” from the raw data.” The reference (number 21) was corrected. The HISAT2 was used. 

Line 104-108: Provide more details on how you performed the DEG analysis.

Eg. Mention any types of normalization done. Did the authors filter out the genes that were low/sparsely expressed?

Response: Done. In the revised version, we detailed the methods in DEG analysis as follow: “To investigate the transcriptomic changes in the gut under different food re-sources, transcripts per million (TPM) expression values were calculated using cufflink [27]. The DESeq2 (http://bioconductor.org/packages/release/bioc/vignettes/Glimma/inst/doc/DESeq2.html) was used to identify the differentially expressed genes with default parameters. Based on these statistical analyses, genes meeting the criteria of P-value < 0.05 and an absolute value of the log2 ratio>1 were deemed as DEGs.”

Line 109-115: Provide detailed methodologies. How did the authors carry out the enrichment analysis? What software was used? What type of analysis was carried out, and what were the background genes for the enrichment analysis?

Response: Done. Thank you for pointing out this. In the revised version, we provide the details in enrichment analyses as follow: “The GO enrichment analyses were performed using TBtools software v1.0697[27]. In this software, enriched P-values were calculated according to one-sided hypergeometric test: , with N represents the number of gene with GO annotation, n represents the number of DEGs in N, M represents the number of genes in each GO term, m represents the number of DEGs in each GO term. The significant pathways were defined based on a corrected P-value ≤ 0.05”.

Line 113 -115 is unnecessary in this section.

Response: Done.

Line 116- 127: How did the authors normalize the expression data, write about any housekeeping genes used. How many biological and technical replicates were used?

Response: Done.

Line 129 – 138: Point out the long and short salivary sheaths in the figure (with arrows) what is the incomplete sheath on leaves and the complete on pods and beans, please elaborate.

Response: Done. Thanks for your suggestion. In the revised version, we displayed the incomplete sheath and complete sheath in yellow and red arrows, respectively.

Figure 1: The healthy and infested samples were incubated both for 4 days in similar conditions except for the presence of insects on the infested samples. The two columns of the SEM are they both are from infested samples? Do you have SEM from healthy samples to compare with?

Response: Done. Thanks for your suggestion. There is no tube-like structure (salivary sheath) in healthy bean, leaf, and pod. The two columns of the SEM are they both are from infested samples.

Line 141-146: Italicize scientific names. Explain the difference between the two columns of the SEM image. Are these SEM images representative of many taken, how many? Give the model of the SEM microscope and the other camera used in imaging in the figure legends.

Response: Done. Thanks for your suggestion. In the revised version, we described in the figure legends that “the short and incomplete salivary sheaths were indicated by yellow arrows, while the long and complete salivary sheaths were indicated by red arrows”. We also described that “For each tissue, more than five biological replicates were performed, and similar results were found”. Also, the model of the SEM microscope and camera used in imaging was described in the figure legend.

Line 156 – 161: G-pod cluster and G-leaf cluster are not separated on PC1 similarly, G-bean and G-pod are not separated on PC2, correct the sentences here.

Response: Done. Sorry about this inaccurate interpretation. We corrected it in the revised version.

Line 164: did the authors mean to write G-leaves or G-leaf instead of G-leave? same in the figure legend.

Response: Done. We corrected it in the revised version.

Line 167- 213 Plot the enrichment analysis results in more meaningful and easier-to-visualize figures showing the enrichment scores and number of genes involved. Figure 4 does not show the enrichment scores. The authors can use heatmaps with a clustering approach to show the expression patterns of the genes in different treatment categories. Tables in the supplementary materials don’t make the picture clear as the authors are presenting it in the text.

Response: Done. Thanks for your suggestion, we have changed this figure to bubble chart. This will improve the presentation of the enrichment results.

Line 215-218: Write it in the method section. How did the authors select these genes? Randomly?

Response: Done. In the revised version, we described that “the top ten DEGs associated with insect detoxification and digestion were selected”,

Line 227- 231: What was the number of biological replicates for the RT-PCR? Is the inconsistency in the expression due to not enough biological and technical replicates?

Response: Done. In the revised version, we described that “Three independent biological replicates with each repeated thrice were performed”. The inconsistency between transcriptome and qRT-PCR might be the different samples used in two methods (we collect new samples in qRT-PCR), and we described that “It is possible that these differences are caused by using different samples in the two methods, leading to variations in gene expression results.”

Line 234-236: The post hoc test used?

Response: Done. In the revised version, we described that “One-way ANOVA test followed by Tukey’s multiple comparisons test” was used to calculated the difference among different groups.

Line 248-249: provide a reference for this line: “The palatability is closely related to the chemical composition of these food resources”.

Response: Done.

Line 249-250: How do the authors relate the shorter salivary sheath with the chemical composition and palatability? Is something missing here? Explain what incomplete and complete salivary sheath is.

Response: Done. In the revised version, we re-wrote this section as: “The palatability is closely related to the chemical composition of these food resources [29, 30]. The gel saliva, which is the main constitute of salivary sheath, is continuous secreted along with insect feeding. The short salivary sheath indicates a shorter probing period during R. pedestris feeding [31]. The salivary sheath on leaves was shorter and incomplete, suggesting lower palatability compared to pods and beans.”

Line 260: Can the authors elaborate, on what they mean by positive change? Do they mean these genes are upregulated while feeding on pods and beans but not while feeding on leaves?

Response: Yes. In the revised version, we rewrote this sentence as “The enhanced expression of these genes while R. pedestris feeding on pods and beans indicates that the insects thrive better on beans and pods compared to soybean leaves, consistent with previous conclusions”.

Line 317: Italicize the scientific name Helicoverpa armigera. Check the entire references for similar mistakes. Eg: Line 326, 328, 330. Check for inconsistencies on the formatting of the references.

Response: Done. Thanks for your suggestions! We have modified this reference and checked other references.

Comments on the Quality of English Language

Writing is understandable. Find my comments above.

Response: Thank you for reviewing our manuscript. We carefully check the English language in the revised version.

Round 2

Reviewer 3 Report

Thanks for addressing my concerns. It looks a lot better now.